# Size-Dependent Distribution of Patient-Specific Hemodynamic Factors in Unruptured Cerebral Aneurysms Using Computational Fluid Dynamics

**DOI:** 10.3390/diagnostics10020064

**Published:** 2020-01-24

**Authors:** Ui Yun Lee, Gyung Ho Chung, Jinmu Jung, Hyo Sung Kwak

**Affiliations:** 1Division of Mechanical Design Engineering, Chonbuk National University, Jeonju 54896, Korea; euiyun93@naver.com; 2Department of Radiology and Research Institute of Clinical Medicine of Chonbuk National University, Biomedical Research Institute of Chonbuk National University Hospital, Jeonju 54907, Korea; chunggh@jbnu.ac.kr; 3Hemorheology Research Institute, Chonbuk National University, Jeonju 54896, Korea

**Keywords:** aneurysm, computational fluid dynamics, non-Newtonian, shear rate, blood viscosity, wall shear stress

## Abstract

Purpose: To analyze size-dependent hemodynamic factors [velocity, shear rate, blood viscosity, wall shear stress (WSS)] in unruptured cerebral aneurysms using computational fluid dynamics (CFD) based on the measured non-Newtonian model of viscosity. Methods: Twenty-one patients with unruptured aneurysms formed the study cohort. Patient-specific geometric models were reconstructed for CFD analyses. Aneurysms were divided into small and large groups based on a cutoff size of 5 mm. For comparison between small and large aneurysms, 5 morphologic variables were measured. Patient-specific non-Newtonian blood viscosity was applied for more detailed CFD simulation. Quantitative and qualitative analyses of velocity, shear rate, blood viscosity, and WSS were conducted to compare small and large aneurysms. Results: Complex flow patterns were found in large aneurysms. Large aneurysms had a significantly lower shear rate (235 ± 341 s^−1^) than small aneurysms (915 ± 432 s^−1^) at peak-systole. Two times higher blood viscosity was observed in large aneurysms compared with small aneurysms. Lower WSS was found in large aneurysms (1.38 ± 1.36 Pa) than in small aneurysms (3.53 ± 1.22 Pa). All the differences in hemodynamic factors between small and large aneurysms were statistically significant. Conclusions: Large aneurysms tended to have complex flow patterns, low shear rate, high blood viscosity, and low WSS. The hemodynamic factors that we analyzed might be useful for decision making before surgical treatment of aneurysms.

## 1. Introduction

Detection of asymptomatic unruptured aneurysms has become feasible with the improvements in neuroimaging methods [1]. After detection of unruptured aneurysms, appropriate treatment for rupture prevention is needed because the discovered unruptured aneurysms are not free from rupture risk [2]. The most widely used significant risk factor for rupture is aneurysm size [3,4], as a large aneurysm is more likely to rupture [5,6]. As the size of the aneurysm increases, blood flow patterns change and the region of low wall shear stress becomes larger [7].

The risk of aneurysm rupture can also be predicted using hemodynamic factors [2]. Since aneurysm size alone might be inappropriate to determine rupture risk of unruptured aneurysm, hemodynamic factors have been assessed using computational fluid dynamics (CFD) [4,5,8]. Several CFD studies have assumed that blood is a Newtonian model with constant viscosity [9,10]. Although use of the Newtonian model reduces computational time and simplifies simulation, it might not be suitable for aneurysm research due to large variation in shear rate [11].

One of the important hemodynamic factors, shear rate, is defined as the rate of change of velocity per distance. Previous studies have reported that the shear rate is dependent upon the geometry and size of the aneurysm. In the dome of the aneurysm, an almost zero shear rate can be observed, but regions of high shear rates also exist [12]. Therefore, a non-Newtonian model reflecting the shear-thinning characteristics of blood flow and the changing viscosity of blood depending on the shear rate should be applied in CFD simulations of aneurysms to study more precisely the large variation in shear rate in the aneurysm dome according to aneurysm size [13,14,15]. Furthermore, shear stress can be calculated by multiplying the viscosity by the shear rate. The method for calculating shear stress is different in Newtonian and non-Newtonian models. To obtain patient-specific shear stress, a non-Newtonian model might be appropriate.

Thus, the purpose of this study was to conduct CFD simulations using patient-specific measurements of non-Newtonian viscosity to accurately assess size-dependent hemodynamic factors within the dome of an unruptured aneurysm. Morphological variables were measured. Quantitative and qualitative comparison of the shear rate, blood viscosity, and wall shear stress (WSS) was carried out between small and large aneurysms.

## 2. Materials and Methods

### 2.1. Patient Selection

This study was approved by our Institutional Review Board, which waived informed consent. Twenty-one patients (8 men and 13 women) with unruptured cerebral aneurysms were enrolled to study size-dependent distributions of hemodynamic factors. All patients were diagnosed with three-dimensional rotational angiography (3DRA). They underwent venous sampling before infusion with physiologic (0.9%) saline when they visited our hospital.

The common locations of cerebral aneurysms in the anterior circulation were identified: 5 aneurysms at an anterior communicating artery, 10 aneurysms at a posterior communicating artery, and 6 aneurysms at a middle cerebral artery. There are various cutoff values (5, 7, and 10 mm) for small and large aneurysms, and we used the cutoff value of 5 mm for this study. Patients with an aneurysm size ≤5 mm and >5 mm were divided into “small” and “large” groups, respectively [6]. The mean age of patients in small aneurysm and large aneurysm groups was 58.1 ± 12.6 years and 61.2 ± 10.5 years, respectively.

### 2.2. Preparation of a 3D Aneurysm Model

All source images were obtained using 3DRA (Axiom Artis dBA; Siemens Medical Solutions, Erlangen, Germany) at total 126 frame during 8.34 s as 1.5° rotation interval and 1024 × 1024 matrix. The obtained 3DRA data was saved in Digital Imaging and Communications in Medicine (DICOM) format for each patient. Using Materialise Mimics v20.0 (Materialise NV, Leuven, Belgium), the obtained data in 2D DICOM format were converted into a patient-specific 3D model in STL format. By controlling and regulating the threshold method, cropping operation, and edit mask in 3D tools, a 3D model was reconstructed for CFD analyses. Unnecessary branches in the 3D aneurysm model were removed before morphology measurements.

### 2.3. Measurement of Morphological Variables

The height, width, and ostium diameter were measured for each patient. The detailed information for illustrating the measurement method is shown in Figure 1. Aneurysm height was obtained by measuring the maximum distance from the neck to the tip of the aneurysm dome, and was denoted as the aneurysm size [6]. The width of the aneurysm was defined as the longest diameter orthogonal to the height of the aneurysm [16]. Using the cross-sectional plane of the aneurysm neck, the maximum diameter of the plane was defined as the ostium diameter [8]. Using the cross-sectional plane of the ostium, the cross-sectional area of the ostium was calculated. The surface area of the aneurysm was obtained using the reconstructed aneurysm dome.

### 2.4. Measurements of Whole Blood Viscosity

After venous sampling of all patients, whole blood viscosity was measured using a scanning capillary tube viscometer (Rheovis-01; Biorheologics Co., Ltd., Jeonju, South Korea). Values for whole blood viscosity at shear rate range from 1 s^−1^ to 1000 s^−1^ were obtained automatically as shown in Figure 2 [17]. Using the data for profiles of whole blood viscosity, the Casson constant and yield stress were applied to properties of non-Newtonian flow in CFD simulations for each patient.

### 2.5. Blood Flow Modeling

The reconstructed 3D model for each patient was imported into COMSOL Multiphysics v5.2a (Comsol, Burlington, MA, USA) for CFD analyses. Fully-coupled method and iterative solver (generalized minimal residual algorithm (GMRES)) were applied for numerical calculation. We set 0.01 as relative tolerance and 50 as maximum number of iteration. The continuity equation and Navier–Stokes equation were used to carry out simulations of blood flow (Equations (1) and (2), respectively). In both equations, flow velocity was represented as *u.* In Equation (2), ρ indicates the density, $\frac{\partial u}{\partial t}$ denotes the rate of change of velocity with time, *p* and μ represent the pressure and fluid viscosity, respectively [9,18]:(1)∇·u=0,
(2)$$\rho \left(\frac{\partial u}{\partial t}\;+\;u\cdot \nabla u\right)\;\;=\;-\nabla p+\mu \nabla^{2}u.$$

Blood flow was considered to be a laminar and incompressible non-Newtonian fluid [19]. For blood viscosity, the measured patient-specific Casson constant and yield stress were applied for CFD analyses:(3)$$\begin{array}{c} \sqrt{\tau }\;=\sqrt{\tau_{y}}+\;\sqrt{k}\sqrt{\dot{\gamma }}\;\;\mathrm{when}\;\tau >{\;\tau }_{y}\\ \dot{\gamma }=0\;\mathrm{when}\;\tau <{\;\tau }_{y} \end{array}$$

In the Casson model (Equation (3)), τ denotes WSS by blood flow, and $\tau_{y}$ indicates the yield stress. k is a Casson constant, and $\dot{\gamma }$ represents the shear rate [20]. The wall was treated as a no-slip condition and rigid due to a lack of information on the thickness and viscoelastic properties of the aneurysm [7,21]. Using the published average flow rate of 2.6 mL/s and patient-specific inlet area, the blood-flow velocity for each patient was calculated and applied to the inlet boundary condition [22,23]. Traction-free boundary condition was implemented at the outlet [24]. The CFD study was conducted for 4 cardiac cycles, and the data were taken from the second cycle of flow simulation [25].

### 2.6. Hemodynamic Factors

Based on CFD simulations, the following calculated hemodynamic factors were obtained: average shear rate; average blood viscosity; minimal, time-averaged, and maximal WSS. The shear rate is defined as the rate of change of velocity per distance, and this term is crucial for assessment of shear stress [26]. The shear rate and blood viscosity were averaged over the volume of the aneurysm dome. Minimal and maximal WSS were defined as the lowest value and highest value during 1 cardiac cycle, respectively. Time-averaged WSS was calculated by integrating the WSS over the duration of one cardiac cycle:(4)$$WSS=\frac{1}{T}\int_{0}^{T}\left|WSS_{i}\right|dt,$$
where *WSS* is the wall shear stress, and *WSS_i_* is the instantaneous wall shear stress, and *T* denotes the duration of one cardiac cycle [1,4]. The minimal, maximal, and time-averaged WSS values were calculated over the surface of the aneurysm dome.

### 2.7. Flow Patterns

Flow patterns between small and large aneurysms were analyzed. The flow pattern can be determined by flow complexity and inflow jets. Flow complexity is divided into “simple” and “complex” depending on the number of flow recirculation areas. Simple flow has 1 recirculation area, and complex flow has 2 or more recirculation areas [27]. The inflow jet can be classified into “diffused” and “concentrated” according to the characteristics of inflow into the aneurysm dome. Diffused inflow flows along the wall of the aneurysm dome. A concentrated inflow jet flows toward the aneurysm tip [25].

### 2.8. Statistical Analyses

All the obtained data for small and large aneurysms were expressed with mean and standard deviation value (mean ± SD). Statistical analyses were undertaken using PASW v18 (IBM, Armonk, NY, USA). To assess significant differences between small and large aneurysm groups, the independent sample *t*-test (for data with a normal distribution) or Mann–Whitney rank sum test (for data with a non-normal distribution) were used. *p* < 0.05 was considered as statistically significant.

## 3. Results

### 3.1. Comparison of Morphological Variables

The average height (aneurysm size), width, ostium diameter, cross-sectional area of the ostium, and surface area of small and large aneurysms are shown in Table 1. Overall, all the morphological features of large aneurysms were larger than those of small aneurysms. Differences in all morphologic variables between small and large aneurysms were statistically significant (Table 1).

### 3.2. Distribution of Velocity

The distributions of blood flow velocity are shown in Figure 3. For comparison between small and large aneurysms, 2 representative cases were used for each aneurysm group: cases A and B for small aneurysms and cases C and D for large aneurysms. Blood flow velocity at peak-systole and end-diastole are shown with the same scale bar (0–0.6 m/s). In the scale bar, red indicates a higher level of blood flow velocity (>0.6 m/s), whereas blue is for flow velocity <0.1 m/s. In cases A and B at peak-systole, high blood flow velocity was observed in the aneurysm dome. In contrast, markedly low blood flow velocity appeared at the aneurysm dome in cases C and D.

Blood flow patterns (flow complexity and inflow jet) were compared between small and large aneurysms, and the recirculation zone was marked by arrows. In small aneurysms, mostly simple flow and diffused inflow were observed. For example, in case B, one recirculation zone was found, and blood flowed along with the wall of the aneurysm dome. In large aneurysms, complex blood flow and diffused inflow were observed. For example, in case D, 2 recirculation zones were observed in the aneurysm dome with diffused inflow.

### 3.3. Shear Rate

Distributions of shear rate at peak-systole and end-diastole are shown in Figure 4. The same scale bar was applied to small and large aneurysms (range, 0–1000 s^−1^). Regardless of aneurysm size, a higher shear rate was found in the parent artery. Due to an increase in aneurysm area, a reduced range of shear rate appeared in the aneurysm dome (1–200 s^−1^). In small aneurysms, a low shear rate was observed at the aneurysm tip, whereas an enlarged region of decreased shear rate was found in the dome of large aneurysms at peak-systole. At end-diastole, a region of higher shear rate was observed around the aneurysm neck for all small and large aneurysms. Compared with the shear rate at peak-systole, a lower shear rate was observed in the parent artery at end-diastole. In large aneurysms, most of the dome showed a lower shear rate (1–100 s^−1^).

The average shear rate at peak-systole and end-diastole was calculated in small and large aneurysms (Table 2). Overall, as the size of the aneurysm increased, a low value of average shear rate was noted. In small aneurysms, the average shear rate at peak-systole was 915 ± 432 s^−1^, whereas a lower average shear rate value at peak-systole was observed in large aneurysms (235 ± 341 s^−1^), and this difference was significant (*p* = 0.008). Compared with the average shear rate of large aneurysms at end-diastole (112 ± 123 s^−1^), 3 times higher average shear rate of small aneurysms was found (339 ± 258 s^−1^), and this difference was significant (*p* = 0.011).

### 3.4. Distributions of Whole Blood Viscosity

Distributions of whole blood viscosity between small and large aneurysms are shown in Figure 5. Higher distributions of blood viscosity were observed at locations where the distributed shear rate was low (reduced from 1 to 100 s^−1^) (Figure 5). In small aneurysms, maximal blood viscosity was shown only at the aneurysm tip. In contrast, as the size increased, a larger area of maximal blood viscosity was found at the aneurysm dome. At end-diastole, a higher value for blood viscosity and a widened area of high blood viscosity appeared in large aneurysms.

Comparison of blood viscosity between small and large aneurysms is shown in Table 3. In large aneurysms, the average viscosity at peak-systole and end-diastole was greater than that in small aneurysms. Large aneurysms had 63.1% higher average blood viscosity than small aneurysms at peak-systole (small: 3.8 ± 0.8 cP; large: 6.2 ± 1.1 cP). At end-diastole, the average blood viscosity of large aneurysms was 1.4 times higher than that of small aneurysms (small: 4.3 ± 0.8 cP; large: 6.1 ± 1.0 cP). The difference in blood viscosity between small and large aneurysms was significant at peak-systole (*p* = 0.001) and end-diastole (*p* = 0.006).

### 3.5. WSS

Distributions of WSS varied depending on aneurysm size (Figure 6). The scale bar was set from 0 to 1 Pa, and was applied to the 2 groups for comparison. In all representative cases, the site of maximal WSS appeared at the aneurysm neck. Moreover, moderately high WSS was found at the parent artery of aneurysms. A region of markedly low WSS was observed at the aneurysm tip, where the shear rate became lower from 1 to 100 s^−1^. The minimum and maximum values of WSS are represented by arrows in Figure 6. In small aneurysms, a small region of low WSS was found at the tip, whereas a large area with low WSS was observed at the dome of large aneurysms. As the aneurysm size increased, the area of low WSS enlarged. At end-diastole, the region of low WSS was widened compared with that at peak-systole, and large aneurysms had wider regions with reduced WSS than that of small aneurysms.

The average minimal, time-averaged, and maximal WSS were calculated at the surface of the aneurysm dome. Overall, small aneurysms tended to have higher WSS than large aneurysms (Table 4). The average minimal WSS for small (2.34 ± 0.77 Pa) and large (1.05 ± 0.97 Pa) aneurysms was significantly different (*p* = 0.011). The mean time-averaged WSS on the aneurysm was 3.53 ± 1.22 Pa for small aneurysms and 1.38 ± 1.36 Pa for large aneurysms, and this difference was significant (*p* = 0.004). In addition, the average maximal WSS was 9.90 ± 3.92 Pa for small aneurysms and 2.90 ± 3.21 Pa for large aneurysms, which was 3.4 times greater, and the difference was significant (*p* = 0.004).

## 4. Discussion

In this study, measured non-Newtonian viscosity was used for a patient-specific CFD study to analyze hemodynamic factors such as velocity, shear rate, blood viscosity, and WSS according to aneurysm size. Based on size-dependent CFD analysis of aneurysms, we were able to identify the different hemodynamic characteristics of small and large aneurysms. Large aneurysms tended to have significantly lower velocity, shear rate, and WSS than those of small aneurysms. In addition, large aneurysms had significantly greater area with increased blood viscosity compared with those of small aneurysms.

### 4.1. Flow Velocity in Small and Large Aneurysms

Cebral et al. [25] reported that complex flow patterns and concentrated inflow jets were associated significantly with aneurysm rupture, whereas simple flow patterns and diffused inflow jets were found commonly in unruptured aneurysms. Xiang et al. [4] showed that, compared with unruptured aneurysms, a complex flow pattern appeared mainly in ruptured aneurysms.

In the present study, 21 patients with unruptured aneurysms were assessed. A diffused inflow jet was observed in all unruptured cases, but flow complexity was different according to size (Figure 3). As the size of aneurysms increased, more recirculation zones were found. According to Cebral et al. [25], complex flow patterns may be natural in unruptured aneurysms. Because unruptured aneurysms have a possibility of becoming ruptured as time goes, unruptured aneurysms may have many of the characteristics of ruptured aneurysms. In addition, complex flow patterns have been known to promote infiltration of inflammatory cells at the aneurysm wall, which increases the risk of aneurysm rupture [4].

### 4.2. Shear Rate and Blood Viscosity

CFD studies have been undertaken based on the assumption of Newtonian blood flow [9,10,28], but this assumption has several limitations. Xiang et al. [11] showed that increased blood viscosity was not found in regions of low shear rate using the Newtonian model. Moreover, the Newtonian model might overestimate the shear rate and WSS, which could lead to inadequate prediction of aneurysm rupture. Xiang et al. [11] also reported that assumption of the Newtonian model hindered study of blood flow patterns in aneurysms. To analyze blood flow accurately, a shear-thinning, non-Newtonian model might be used to conduct CFD analyses [29].

In the present study, hemodynamic factors were analyzed with measured non-Newtonian model for more precise results. As a result, a zone of decreased shear rate was captured at the aneurysm dome (shear rate from 1 to 200 s^−1^; Figure 4). Also, increased blood viscosity was observed in the region where the distributed shear rate decreased (Figure 5). Flow velocity in aneurysms decreased due to an increase in the area of the aneurysm, which resulted in greater reduction in the shear rate in large aneurysms.

### 4.3. High WSS or Low WSS

Although pathogenesis of aneurysm formation is not a solved problem, WSS has fundamental roles in the formation, growth, and rupture of cerebral aneurysms [4,10,30]. Whether aneurysm rupture is associated with high or low WSS is controversial [31,32]. Cebral et al. [33] reported that high and low WSS were associated with the growth and rupture of cerebral aneurysms, respectively.

The layer of endothelial cells in blood vessels can be damaged by high WSS [34]. Aneurysm initiation involves several steps: mural cells secrete matrix metalloproteinases; the internal elastic lamina is damaged [35]; apoptosis occurs; the media layer becomes thinner; and a bulge is formed [36]. During aneurysm initiation, infiltration of inflammatory cells is not observed [37].

When impinging blood flow remains with excessively high WSS after bulge formation, the mechanisms involved in aneurysm initiation continue to occur [36]. Degradation and death of mural cells occur because of decreased cell anchorage [38]. Aneurysms with high WSS are acellular, transparent, and small. Conversely, if the aneurysmal bulge becomes suitably large, low WSS appears [36]. Low WSS progressively promotes the biologic mechanisms related to remodeling of damaged walls [38,39]. Pro-inflammatory endothelial cells spend long periods of time in blood and promote leukocyte transmigration during dysfunctional remodeling of the wall. Many inflammatory cells are observed in large aneurysms [40].

We showed that high WSS was observed in small aneurysms, and large aneurysms had low WSS. Shojima et al. [9] reported that a suitable WSS for maintaining the vascular structure was ~2.0 Pa, and that WSS <1.5 Pa was associated with degradation of endothelial cells through apoptosis. In the present study, a time-averaged value of 1.38 ± 1.36 Pa was observed in large aneurysms (Table 4). In addition, the increased amount of time spent by leukocytes in blood in large aneurysms might have caused high blood viscosity in large aneurysms (Table 3). A consensus as to which level of WSS (high or low) is closely associated with aneurysm rupture is not available because high and low WSS have critical roles in aneurysm formation, and the magnitude of WSS is different depending on aneurysm size [22,41].

### 4.4. Limitations of This Study

Our study had 3 main limitations. First, although the results were statistically significant, our study population was small. Only 21 patients with unruptured aneurysms were enrolled due to the difficulty of collecting venous blood. Not all patients with aneurysms who visited our hospital participated in our study because they were emergency cases or they had been transferred from other hospitals and saline injection was in progress. Fortunately, the 21 patients who participated in our study completed venous sampling before saline injection.

Second, converting the obtained source image into a model of a 3D aneurysm could have resulted in errors. Reconstructed models of 3D aneurysms can be affected (at least in part) by inaccurate thresholding or removal of unnecessary branches of aneurysms. Therefore, we attempted to obtain a source image of the aneurysm of optimal quality using 3DRA (not using a 2D image obtained through computed tomography or magnetic resonance imaging) to minimize these potential errors.

The final limitation was that we applied commonly used boundary conditions due to the difficulty of measuring patient-specific data. The inlet boundary condition was calculated based on the flow rate in the literature. We also used the wall properties of “rigid” and “no-slip” condition. For more accurate analyses, measurement of patient-specific blood viscosity was undertaken instead of applying a Newtonian model for blood. In future studies, we will attempt to measure the inlet velocity using phase-contrast magnetic resonance imaging.

## 5. Conclusions

By applying measured patient-specific non-Newtonian viscosity to CFD analysis, hemodynamic factors according to aneurysm size could be thoroughly compared. Large aneurysms had lower velocity, shear rate, and WSS in the aneurysm dome. Diffused inflow jets were found in small and large aneurysms. Small aneurysms had simple flow patterns, whereas large aneurysms had complex flow patterns. Blood viscosity was higher in the domes of large aneurysms. All analyzed hemodynamic factors showed statistically significant differences between small and large aneurysms. The hemodynamic factors that we analyzed could help to predict the rupture risk of aneurysms and aid decision-making before surgical planning.

## Figures and Tables

**Figure 1 diagnostics-10-00064-f001:**
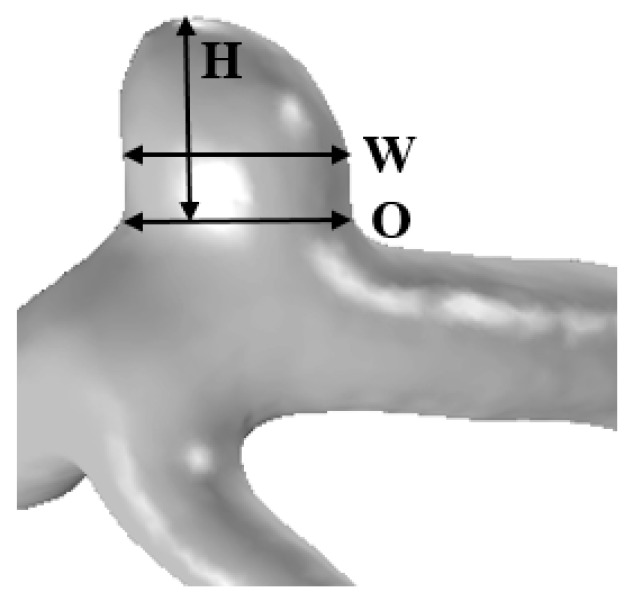
The reconstructed 3D patient-specific geometry of aneurysm illustrating the measurement method: height (H), width (W), and ostium (O).

**Figure 2 diagnostics-10-00064-f002:**
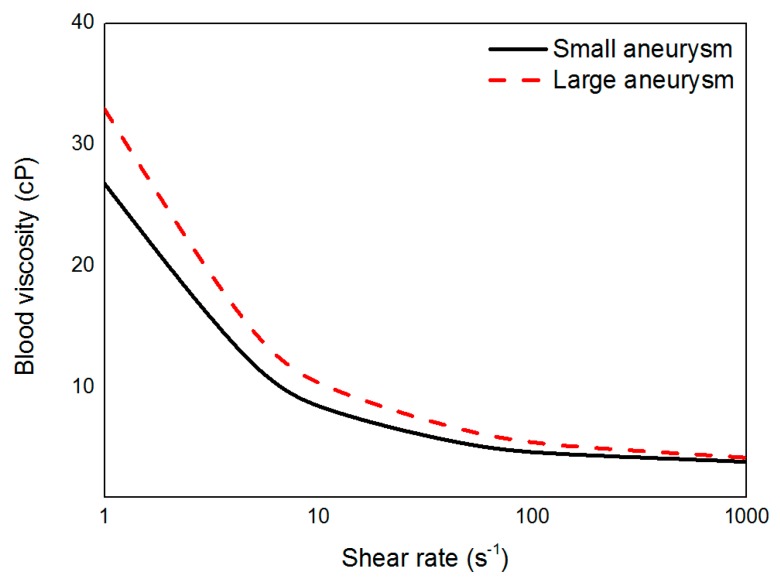
Measured blood viscosity profiles of small and large aneurysms according to shear rate.

**Figure 3 diagnostics-10-00064-f003:**
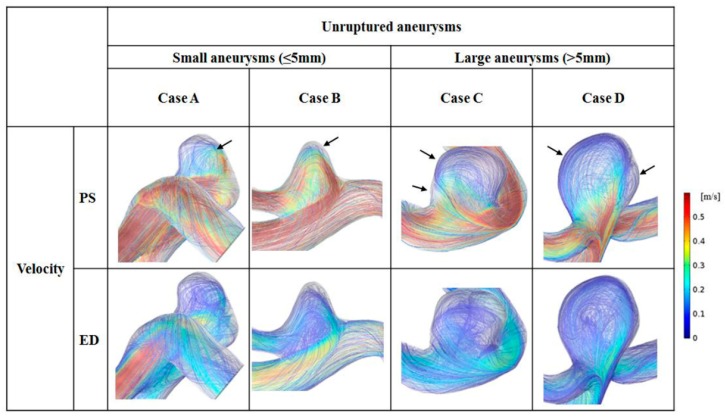
Flow patterns for 2 representative small and 2 representative large unruptured aneurysms at peak-systole (PS) and end-diastole (ED). Black arrows indicate the recirculation zones in the aneurysm dome.

**Figure 4 diagnostics-10-00064-f004:**
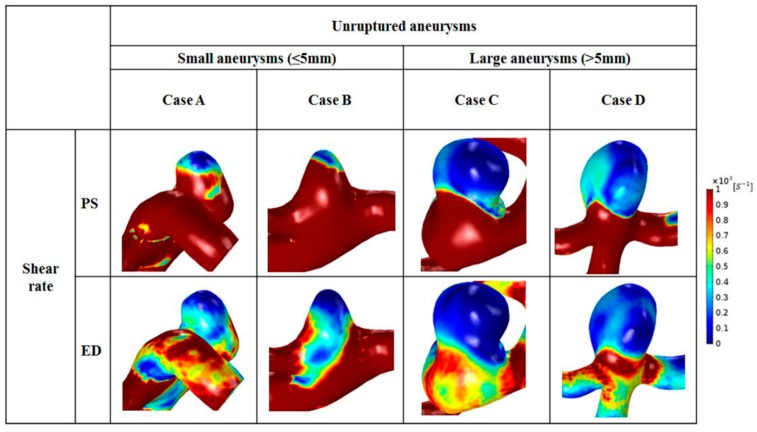
Distributions of shear rate for representative cases for small (cases **A** and **B**) and large (cases **C** and **D**) aneurysms during the cardiac cycle.

**Figure 5 diagnostics-10-00064-f005:**
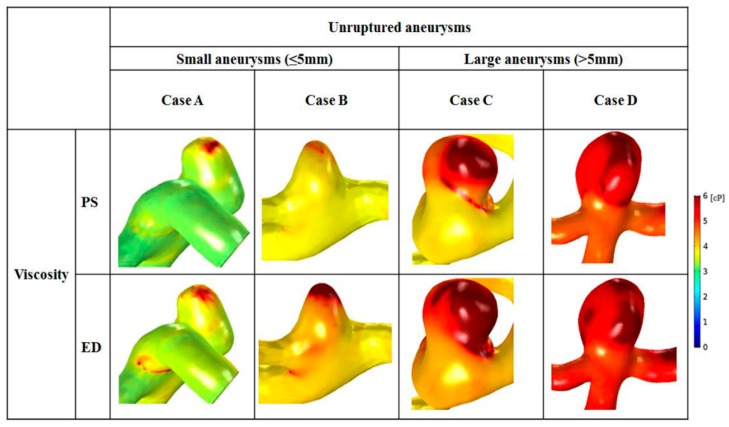
Distributions of whole blood viscosity are shown with representative cases for small (cases **A** and **B**) and large (cases **C** and **D**) aneurysms. An enlarged area with increased blood viscosity was observed in cases C and D.

**Figure 6 diagnostics-10-00064-f006:**
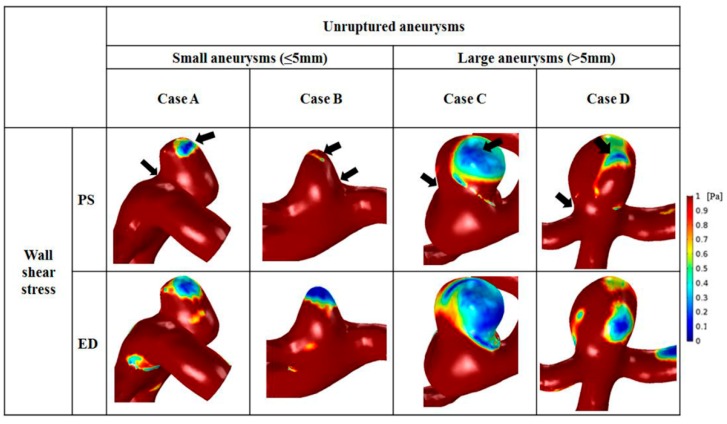
Comparison of wall shear stress between small and large aneurysms. The black arrows indicate that the area with the highest wall shear stress was at the aneurysm neck, and that the area with lowest wall shear stress was at the aneurysm tip.

**Table 1 diagnostics-10-00064-t001:** Morphological variables in unruptured aneurysms.

	Small Aneurysms (≤5 mm)	Large Aneurysms (>5 mm)	*p*
	Mean ± SD (*n* = 16)	Mean ± SD (*n* = 5)	
Age (years)	58.1 ± 12.6	61.2 ± 10.5	0.632
Height (mm)	3.4 ± 0.8	6.2 ± 2.4	<0.001 *
Width (mm)	4.5 ± 1.5	7.4 ± 2.5	0.005 *
Ostium diameter (mm)	4.5 ± 0.8	5.9 ± 1.1	0.005 *
Cross-sectional area of ostium (mm^2^)	12.8 ± 4.1	18.7 ± 5.1	0.014 *
Surface area of aneurysm (mm^2^)	57.7 ± 24.2	165.5 ± 118.5	0.003 *

* indicates *p* < 0.05; SD: standard deviation.

**Table 2 diagnostics-10-00064-t002:** Comparison of shear rate between small and large aneurysms.

	Unruptured Aneurysms		
	Small Aneurysms (≤5 mm)	Large Aneurysms (>5 mm)	*p*
	Mean ± SD (*n* = 16)	Mean ± SD (*n* = 5)	
Average shear rate at PS (s^−1^)	915 ± 432	235 ± 341	0.008 *
Average shear rate at ED (s^−1^)	339 ± 258	112 ± 123	0.011 *

* indicates *p* < 0.05; SD: standard deviation; PS: peak-systole; ED: end-diastole.

**Table 3 diagnostics-10-00064-t003:** Comparison of blood viscosity between small and large aneurysms.

	Unruptured Aneurysms		
	Small Aneurysms (≤5 mm)	Large Aneurysms (>5 mm)	*p*
	Mean ± SD (*n* = 16)	Mean ± SD (*n* = 5)	
Average viscosity at PS (cP)	3.8 ± 0.8	6.2 ± 1.1	0.001 *
Average viscosity at ED (cP)	4.3 ± 0.8	6.1 ± 1.0	0.006 *

* indicates *p* < 0.05; SD: standard deviation; PS: peak-systole; ED: end-diastole.

**Table 4 diagnostics-10-00064-t004:** Comparison of wall shear stress between small and large aneurysms.

	Unruptured Aneurysms		
	Small Aneurysms (≤5 mm)	Large Aneurysms (>5 mm)	*p*
	Mean ± SD (*n* = 16)	Mean ± SD (*n* = 5)	
Minimal wall shear stress (Pa)	2.34 ± 0.77	1.05 ± 0.97	0.011 *
Time-averaged wall shear stress (Pa)	3.53 ± 1.22	1.38 ± 1.36	0.004 *
Maximal wall shear stress (Pa)	9.90 ± 3.92	2.90 ± 3.21	0.004 *

* indicates *p* < 0.05; SD: standard deviation.

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
