# Peer review of "Size-Dependent Distribution of Patient-Specific Hemodynamic Factors in Unruptured Cerebral Aneurysms Using Computational Fluid Dynamics"

_diagnostics, 2020, doi:10.3390/diagnostics10020064_

Round 1

Reviewer 1 Report

The paper by Lee at al. considers a patient specific analsis of cerebral aneurysms and is based on cfd simulations. I believe the topic is of interest for the Journal and that the followed approach is currently widely spread in the refernce Community.

Neverthless it needs to be improved before it may be suitable for the pubblication.  I would suggest to address the following issues:

INTRODUCTION

-Provide a wider background in the introduction focussing on the described phenomenon, its physical and medical implications, the main results of the literature in this framework and emphasizing how your outcomes differ from them and also how they may be used in driving the possible surgery treatments;

MATERIAL AND METHODS

-More details are required on the model description, among them:

 add a scheme of what you described in section 2.3 and 2.4;

add details on the model equations and on the numerical model itself, also on the fluid dynamical aspects: justify the use of the considered BCs , what does the'published' flow rate physically represents? the considered Reynolds number....;

RESULTS

In general: it is not clear to me why you consider only 4 cardyac cycles to perform time statistics also. Also, in my view, a multi-component analysis need to be consider in the description of such a problem.

-In the fluid flow description 2D maps (of velocity/vorticity, strain, WSS) have to be added in order to better visualize the correlation between the geometry and the emodynamics; I feel the flow patterns description is a bit too qualitative;

-factors like OSI and RRT (see i.e. Appl. Sci. 20199(7), 1341) have not considered: do you think they may influence your study?

CONCLUSIONS

Need to be updated after having carefully considered the previous issues.

Reviewer 2 Report

In principle, the manuscript looks good.

There are two major aspects, which should be considered by the authors:

1) The design of the work is not clear. In the introduction the authors mention, that significant risk factor is the size of an aneurysm (page 1, line 39). And the decision for treating small aneurysms is a dilemma (page 1, line 42). So, the big aneurysms are not a problem for decision making. The problem is to identify, which of the small aneurysms has increased potential risk to rupture. But then, the authors state their goal as a comparison of big and small aneurysms, which is not logical!

2) There is a tremendous number of works devoted to the problem of aneurysms rupture. The authors should clearly indicate, what are the advantages and novelty of their work.

Round 2

Reviewer 1 Report

I noticed that the the Authors addressed most of the issues raised by both the Rewievers. Neverthless thera are still several things to be improved before the paper can be suitable for the publication.

Point 1:
The introduction has been improved and is now more complete but it still shows some lacks. Among them it is not clear if big aneurysm are not a problem as far as treatment decision is concerned, why are they investigated? Also the novelty of the work is still undefined. As a matter of fact, to investigating the possible correlations between geometrical and hemodynamical factors is  widely used in appraoching these kinds of problems. So, what's new?

Point 2 : OK.

Point 3 : My comments was  mostly on the adopted numerical scheme that should have been better described (instead of reporting the very well known N/S equations).

Point 4 : I am still convinced on the fact that 4 cardyac cicles represent are too few samples to perform statistics....

Point 5: it is not clear too me. If you are able to plot 3D maps you should also have the corresponding dataset (i.e. speaking about velocity, for any x,y,z in your domain you should have the corresponding components u,v,w). So why it is not possible to extract 2D information from these datasets?

And about flow complexity, I would absolutely avoid to associate this very crucial concept to a trivial feature as the presence of one or two recirculations. It is more convenient to speak about different flow patterns but not of different level of complexity.

Point 6: Ok.

Point 7: Need to be updated in consideration of the previous points.

Reviewer 2 Report

I'm satisfied with the answer to point 1, but the novelty of the work (point 2) still hidden. From the text, it is not clear, what is new/unique key in this work? There are a lot of works in the same scope, e.g. DOI 10.1016/j.jbiomech.2013.09.004i, 10.1115/1.3148470, 10.3340/jkns.2017.0314 10.1088/1742-6596/477/1/012001 and many many others ...

Round 3

Reviewer 1 Report

The revised manuscript appears now to be suitable for the pubblication